# Identification of a Small Molecule Compound Active against Antibiotic-Tolerant *Staphylococcus aureus* by Boosting ATP Synthesis

**DOI:** 10.3390/ijms24076242

**Published:** 2023-03-26

**Authors:** Ho-Ting-Venice Iu, Pak-Ming Fong, Hin-Cheung-Bill Yam, Peng Gao, Bingpeng Yan, Pok-Man Lai, Victor-Yat-Man Tang, Ka-Ho Li, Chi-Wang Ma, King-Hei-Kenneth Ng, Kong-Hung Sze, Dan Yang, Julian Davies, Richard-Yi-Tsun Kao

**Affiliations:** 1Department of Microbiology, School of Clinical Medicine, Li Ka Shing Faculty of Medicine, The University of Hong Kong, Hong Kong, China; 2Morningside Laboratory for Chemical Biology, Department of Chemistry, The University of Hong Kong, Hong Kong, China; 3Department of Microbiology and Immunology, The University of British Columbia, Vancouver, BC V6T 1Z3, Canada; 4State Key Laboratory of Emerging Infectious Diseases and the Research Centre of Infection and Immunology, Li Ka Shing Faculty of Medicine, The University of Hong Kong, Hong Kong, China

**Keywords:** MRSA, antibiotic tolerance, adjuvants, persistent infections, persister

## Abstract

Antibiotic tolerance poses a threat to current antimicrobial armamentarium. Bacteria at a tolerant state survive in the presence of antibiotic treatment and account for persistence, relapse and recalcitrance of infections. Antibiotic treatment failure may occur due to antibiotic tolerance. Persistent infections are difficult to treat and are often associated with poor prognosis, imposing an enormous burden on the healthcare system. Effective strategies targeting antibiotic-tolerant bacteria are therefore highly warranted. In this study, small molecule compound SA-558 was identified to be effective against *Staphylococcus aureus* that are tolerant to being killed by conventional antibiotics. SA-558 mediated electroneutral transport across the membrane and led to increased ATP and ROS generation, resulting in a reduction of the population of antibiotic-tolerant bacteria. In a murine chronic infection model, of which vancomycin treatment failed, we demonstrated that SA-558 alone and in combination with vancomycin caused significant reduction of MRSA abundance. Our results indicate that SA-558 monotherapy or combinatorial therapy with vancomycin is an option for managing persistent *S. aureus* bacteremia infection and corroborate that bacterial metabolism is an important target for counteracting antibiotic tolerance.

## 1. Introduction

Persisters were first described by Joseph Bigger in 1944, when he found a population of *S. aureus* surviving the treatment of a high concentration of penicillin [1]. Studies [2,3,4,5,6] showed that *S. aureus* in a dormant state exhibits a high level of tolerance toward antibiotic killing. In the setting of a laboratory, the proportion of antibiotic-tolerant cells in a *S. aureus* population varies with the growth phase [4,7]. The entire population of stationary-phase bacteria survive in the presence of high concentrations of bactericidal antibiotics [2,3,5,7]. This is now termed as antibiotic “tolerance” [8]. In contrast, *S. aureus* in the lag and early exponential phase are susceptible to being killed by antibiotics [7]. The survivors after antibiotic killing are defined as “persisters” [8].

Antibiotic tolerance can be caused by a number of factors, namely reduced metabolism [9] and failure to generate ROS [10]. It is a consequence of the biosynthetic processes that traditional antibiotics target for killing being significantly reduced [11] or antibiotic uptake being arrested as a result of low-energy state [12]. Clinically, antibiotic tolerance is associated with relapse and recalcitrance of infections [13] and complicates antibiotic treatment strategies for infections [14]. Treatments for persistent staphylococcal infections could range from one week to over three months and require a combination of multiple antibiotics [15]. The longer treatment course and the use of multiple antibiotics increase antibiotic exposure to bacteria and amplify the emergence of multidrug resistance [16,17]. Moreover, persistent infections often result in poor prognosis. Patients with *S. aureus* persistent bacteremia had significantly higher mortality rate than those with non-persistent ones [18]. Prolonged hospitalization and increased medical attention are required for persistent infections, imposing a greater burden on the healthcare sector [18]. The discovery of safe and efficacious therapeutics targeting antibiotic-tolerant bacteria is urgently warranted. Studies have presented that metabolism plays a central role in bacterial persistence or tolerance and might be attractive in developing therapeutics counteracting antibiotic tolerance [2,6,19,20].

Antibiotic tolerance and persistence are generally characterized by a low-energy state, and increasing energy production would activate and sensitize the antibiotic-tolerant bacteria to antibiotic killing [9,11]. Since gentamicin uptake is energy-driven, a compound that increases its accumulation might be effective in targeting antibiotic-tolerant bacteria [2,20,21]. In the compound screening, we aim to identify a small molecule compound which is capable of increasing the availability of gentamicin in bacterial cells. A small molecule library composed of 50,240 compounds was screened, and SA-558 was found to be active against antibiotic-tolerant *S. aureus*, in vitro and in vivo. SA-558 mediates electroneutral transport across the membrane and increases energy production and subsequently reduces the population of antibiotic-tolerant bacteria by high ROS generation. It is noticeable that no cytotoxicity of SA-558 was observed in mammalian cell-based assays and in mice following 10-day consecutive high-dose treatments.

## 2. Results

### 2.1. SA-558 Was Identified as an Effective Compound against Antibiotic-Tolerant S. aureus

A chemical library composed of 50,240 diverse small molecules was screened to identify compounds that act against *S. aureus* when they were used in combination with the aminoglycoside antibiotic gentamicin (100 μg/mL) (Appendix A). SA-558 (structure shown in Figure 1a) was selected for further characterization because of its large combinatorial effect with gentamicin and low cytotoxicity level (Appendix A).

After that, time kill kinetics were performed to evaluate its activity toward bacteria from different growth phases. Stationary-phase bacteria are known to be tolerant to being killed by antibiotics [10] while a small proportion of antibiotic-tolerant bacterial cells are present in the exponential phase. In agreement with the screening results, in exponential-phase *S. aureus*, SA-558 on its own had no activity but caused 2-log reduction in bacterial viability when combined with gentamicin (Figure 1b). SA-558, up to 100 μg/mL tested, showed no activity against exponential phase bacteria (Appendix A). As expected, treatments of stationary phase bacteria with different antibiotics alone caused no decrease in viable count, indicating they are tolerant to being killed by antibiotics (Figure 1c,d). In contrast, SA-558 alone caused around 1-log reduction in bacterial viability of stationary phase *S. aureus* (Figure 1d). Two-log reduction in bacterial viability of stationary-phase *S. aureus* was observed when combined with 15 μg/mL of gentamicin (Figure 1d). SA-558’s effect on exponential phase and stationary phase *S. aureus* was further tested in two CA-MRSA strains, which are USA300-3 and LAC, and similar observations were attained in the Mu3 strain (Appendix A). These results indicated that SA-558 alone is preferentially active against stationary phase *S. aureus* that can tolerate antibiotic killing.

### 2.2. SA-558’s Anti-Antibiotic-Tolerant Effect Is Mediated by an Increase in Membrane Potential

By using LC-MS/MS, we demonstrated that SA-558 increased intracellular gentamicin concentration in bacterial cells (Figure 2a). The uptake of aminoglycosides depends on proton motive force (PMF) [2,22], and PMF has two components, which are membrane potential (∆ψ) and proton gradient (∆pH) [23]. Here, we first test the effect of SA-558 on membrane potential in bacteria. We pre-incubated the culture with carbonyl cyanide m-chlorophenyl hydrazone (CCCP), which dissipates proton motive force, before treating with combination of SA-558 and gentamicin. CCCP treatment abolished SA-558 gentamicin-potentiating activity, indicating that the proton motive force is required for SA-558 activity (Figure 2b). We measured the effect of SA-558 on membrane potential with a voltage-sensitive dye, DiBAC4(3). *S. aureus* cells treated with a range of different concentrations of SA-558 showed significant reduction in fluorescence and the reduction in DiBAC4(3) fluorescence, occurring in a concentration-dependent manner (Figure 2c,d). This indicated that SA-558 increased membrane potential. The increase in membrane potential caused by SA-558 can be seen in both exponential phase (Figure 2c) and stationary phase *S. aureus* (Figure 2d). Larger hyperpolarization was observed with stationary phase *S. aureus* (Figure 2d), as compared to exponential phase *S. aureus* (Figure 2c), and this may account for more pronounced SA-558 activity in stationary phase *S. aureus*. Taken together, SA-558 increases membrane potential to increase aminoglycoside uptake.

### 2.3. SA-558 Activity Is Mediated by the Disruption of pH Homeostasis

To test if pH plays a role in the SA-558 mode of action, we tested SA-558 antibacterial activity as a function of media pH. SA-558’s MIC was 1.68 µg/mL when media pH was 5.5, where ∆pH component of the PMF dominated and increased as media pH increased (Figure 3a). SA-558 activity was abolished when media pH was 8.5 (Figure 3a and Appendix A). We further tested SA-558’s effect on intracellular pH in bacteria by monitoring pHluorin fluorescence. Bacteria maintained an intracellular pH at around 8 regardless of external pH (Figure 3b). Intracellular pH decreased in the presence of SA-558, and the extent of the decrease caused by SA-558 varied with external pH (Figure 3b). The effect of SA-558 on intracellular pH was more profound in conditions when external pH ranges from 5.5 to 7.5 (Figure 3b).

We next addressed the question of how SA-558 alters intracellular pH in bacteria. By using an HPTS (8-hydroxypyrene-1,3,6-trisulfonic acid) assay described in [24], we showed that SA-558 equilibrated the intraliposomal pH with external pH without assisting ionophores in a protein-free liposome (Figure 3c). The results indicated that SA-558 may mediate electroneutral transport across the membrane in a manner which is independent of any specific protein target and hinted that SA-558 may dissipate ∆pH of the proton motive force.

### 2.4. SA-558 Disrupts Redox Homeostasis

It is anticipated that the level of ATP would increase following increased membrane potential [25]. Cell Titer Glo assay was employed to measure the intracellular ATP level. Different growth phases were tested. SA-558 was found to increase ATP levels in stationary phase *S. aureus* but not exponential phase *S. aureus* (Figure 4a,b). The increase in ATP level caused by SA-558 was not observed in Vero cells (Figure 4c). This indicated that the effect of SA-558 on ATP level is specific to the stationary phase bacterial cells and may account for SA-558 preferential activity towards antibiotic-tolerant stationary phase *S. aureus*.

Previous studies [26,27] suggested that increasing ATP would lead to redox physiological alteration and contribute to the lethality of bacteria. Following the observation that SA-558 increased ATP levels in stationary phase *S. aureus* (Figure 4b), we hypothesized that SA-558 activity is mediated by ROS production and tested the hypothesis by using ROS indicators. Studies [10,28] showed that bactericidal antibiotics kill bacteria by induction of hydroxyl radicals, and perturbations which interfere with ROS accumulation result in antibiotic tolerance. Hydroxylphenyl fluorescein (HPF), which is oxidized by hydroxyl radicals with high specificity [29], was used to study the effect of SA-558 and/or gentamicin in bacteria. Gentamicin only caused a minor increase in ROS level, which may explain the gentamicin tolerance observed in bacteria (Figure 4d). In contrast, SA-558 significantly increased the ROS level in *S. aureus* cells (Figure 4d). When combined with gentamicin, the ROS level in bacterial cells was further enhanced (Figure 4d). We further tested the effect of SA-558 on ROS production by using a general ROS indicator, H2DCFDA. SA-558, at 26.25 µg/mL, significantly increased the ROS level in stationary phase *S. aureus* cells, and the increase in ROS caused by SA-558 was suppressed by thiourea, an antioxidant (Figure 4e). Furthermore, SA-558’s antibacterial activity in stationary phase *S. aureus* was reduced by the addition of thiourea (Figure 4f). Taken together, SA-558’s activity against antibiotic-tolerant *S. aureus* population is mediated by ROS induction.

### 2.5. SA-558 Exhibited Efficacy in a Murine Persistent Bacteremia Infection Model

To investigate the potential efficacy of SA-558 in vivo, we first established a persistent bacteremia infection model in mice. The mice were intravenously (i.v.) infected with different forms of *S. aureus*: (1) a stationary phase culture, (2) a stationary phase culture treated by ciprofloxacin (4× MIC), and (3) an exponential phase culture. Mice infected with persistent forms (stationary phase and ciprofloxacin-treated stationary phase) develop less severe infection, when compared to exponential phase bacteria. At similar inoculum, no lethality was observed in groups inoculated with (1) the stationary phase culture and (2) the stationary phase culture with ciprofloxacin treatment, while 80% of mice inoculated with (3) the exponential phase culture died (Figure 5a). The effect on body weight was smaller in the group inoculated with the ciprofloxacin-treated stationary phase culture when compared to the stationary phase culture (Figure 5b). As shown in Figure 5c, the overnight culture exposed to ciprofloxacin (4× MIC) treatment has four-fold lower bacterial abundance, as compared to that without ciprofloxacin treatment. As ciprofloxacin treatment selects a proportion of the whole population (stationary phase culture), the stationary phase culture treated by ciprofloxacin might be the most representative form of the cultures for studying persistent infections [7,8,30].

The feasibility of SA-558 as therapeutic for persistent infections was then tested. We first infected the mice with ciprofloxacin-treated stationary phase *S. aureus* ST 239 IIIAH and allowed the infection to develop for 7 days, followed by treatment for 6 days. In our experimental model, vancomycin, at 4.5 mg/kg and 15 mg/kg, did not significantly reduce MRSA abundance (Figure 5d), although the ST 239 IIIAH strain is vancomycin-sensitive (Appendix A), corroborating that bacterial cell in the infection model are persisters. We demonstrated that SA-558, at 2.63 mg/kg, significantly reduced MRSA abundance in kidneys of infected mice (*p* = 0.04), as compared to the vehicle control group (Figure 5d). No bacteria were detected in four out of 17 infected mice treated with 2.63 mg/kg SA-558 (Figure 5d). There was a significant reduction in MRSA abundance in kidneys of the infected mice (*p* = 0.0013), as compared to the vehicle control group, and no bacteria were detected in one-fourth of the infected mice that received combinatorial treatment of SA-558 and 4.5 mg/kg vancomycin (Figure 5d). Compared to 4.5 mg/kg vancomycin alone, a significant reduction in bacterial abundance is also observed in the group treated by a combination of vancomycin and SA-558 (*p* = 0.02). This showed that the combination of SA-558 and vancomycin might be an effective strategy in managing persistent infections that could not be treated by vancomycin alone.

## 3. Discussion

A growing body of research demonstrated close links between bacterial metabolism, proton motive force, and antibiotic tolerance [31]. Traditional antibiotics target energy consuming processes for their bactericidal activity, and reports showed that dysregulation of bioenergetics modulates antibiotic lethality [11,32]. It has been reported that conditions such as low nutrient availability [33], hypoxia [34], and low pH [35] may shift the bacteria to an antibiotic-tolerant state. These conditions lead to reduced carbon metabolism and TCA cycle activity [19,33], subsequently reducing the flow of electron donors to the proton pumps of the electron transport chain, resulting in reduced proton motive force and ATP synthesis [19]. The low ATP level compromised the efficacy of traditional antibiotics [4,9]. Moreover, antibiotic uptake is a result of the dynamic between influx via different processes, for example, diffusion and active transport and efflux pumps [36,37]. It is shown that the heterogeneity of the antibiotic uptake efficiency in a bacterial population is correlated to varying antibiotic efficacy in bacterial cells [38], and reduced antibiotic accumulation is observed in stationary phase bacteria, when compared to exponential phase bacteria [12], accounting for the reduced antibiotic killing in stationary phase bacteria.

Bioenergetics is an important target for developing new therapeutics against antibiotic-tolerant bacteria. Several reports demonstrated metabolite supplementation could restore antibiotic efficacy in antibiotic-tolerant bacteria. [2,39,40]. However, translating these findings to therapeutics could be difficult because the intake of large amounts of metabolites might result in undesirable physiological consequences. Therefore, finding non-metabolite compounds targeting the bioenergetics of bacteria will be a novel therapeutic approach in overcoming antibiotic tolerance. Based on the earlier work by Allison and Collins [2] and Mok et al. [20], we reasoned that the aminoglycoside potentiation phenomenon might be helpful for identifying compounds targeting metabolism to counteract antibiotic tolerance [2,21]. In brief, aminoglycoside uptake depends on the proton motive force, and its efficacy is largely impaired in deenergized bacterial cells such as persisters, and molecules potentiating aminoglycoside via proton motive force stimulation would kill antibiotic-tolerant bacteria [2,21]. In this study, we screened a chemical library for aminoglycoside potentiating activity, and the hits were followed for their activity against antibiotic-tolerant bacteria. By using this approach, we identified a small molecule compound, SA-558, which was effective against antibiotic-tolerant *S. aureus* both in vitro (Figure 1d) and in vivo (Figure 5d). It is worthwhile to note that no cytotoxicity was detected in mammalian cell-based assays (Appendix A), and no adverse effects were observed following 10-day consecutive treatments in mice (Appendix A) with a high dose of SA-558, likely due to the inability of SA-558 to disrupt the cellular ATP level in mammalian cells (Figure 4c).

Mechanistic studies indicated that SA-558 is a membrane-targeting compound. Changes in membrane integrity would affect bacterial homeostasis, leading to fundamental metabolic disorder, for example perturbation of the proton motive force. The proton motive force is essential for bacterial survival and is generated by bacterial transmembrane potential, which is composed of the transmembrane proton gradient (∆pH) and electrical potential (∆ψ) [23]. Disruption of ∆pH would be compensated by increasing ∆ψ [23]. At a low pH, ∆pH predominates, and ∆ψ decreases for compensation [23]. Conversely, ∆ψ predominates at a higher pH [23]. We assayed SA-558 antibacterial activity in different pH levels from pH 5.5 to 8.5 and observed improved activity at a lower pH, of which ∆pH predominates (Figure 3a). This indicated that SA-558 might be targeting the ∆pH component of proton motive force. We further explore how SA-558 alters pH by using pHluorin fluorescence in bacteria and an HPTS assay in protein-free liposome. We found that the intracellular pH decreased in the presence of SA-558, and the effect of SA-558 decreased as external pH increased (Figure 3b). In an HPTS assay, we showed that SA-558 equilibrated internal pH with external pH without assisting ionophores (Figure 3c), indicating that SA-558 mediated electroneutral transport across the membrane [24]. With an influx of protons to bacterial cells resulting in a reduction in the intracellular pH, positively charged species (cations) should be exchanged to maintain electroneutrality. In inductively coupled plasma mass spectrometry (ICP-MS) analysis of intracellular metal ion concentration of the samples, we observed a significant reduction in Mg2+ and K+ concentration in SA-558-treated samples, as compared to the DMSO control (Appendix A). Dissipation of ∆pH would lead to a counter-increase in ∆ψ. To prove this, we measured membrane potential by using voltage-sensitive dye and found that SA-558 increased membrane potential (Figure 2c,d). Consistently, SA-558 effect is antagonized by CCCP, which dissipates the proton motive force. Taken together, SA-558 may act on the ∆pH component of proton motive force. Membrane potential (∆ψ), but not proton gradient (∆pH), is essential for ATP synthesis [41]. Driven by increased membrane potential caused by SA-558, ATP synthesis in bacterial cells increased (Figure 4b). An increased ATP level causes redox-related physiological alteration and contributes to lethality of bacteria [26,27]. Our experimental data showed that SA-558 increased the ATP level (Figure 4b) and the ROS level (Figure 4d) in bacterial cells, which may contribute to SA-558 antibacterial activity. In addition, the uptake of aminoglycosides is driven by membrane potential [42]; thereby, the uptake of gentamicin (a type of aminoglycoside) is increased by SA-558 (Figure 2a), and thus gentamicin killing is increased.

Our results showed that SA-558 was more active against stationary phase *S. aureus*, as compared to exponential phase *S. aureus* (Figure 1d and Figure 4a,b). We observed that SA-558 increased the ATP level in stationary phase *S. aureus* (Figure 4b) but not in the exponential phase *S. aureus* (Figure 4a). Increasing ATP levels may cause redox alteration in bacterial cells and contribute to the lethality of bacteria [26,27]. As expected, we found that SA-558 induced ROS production in bacteria (Figure 4d,e), and SA-558 activity is reduced by thiourea, an ROS scavenger (Figure 4f). This indicated that SA-558 activity is mediated by ROS induction. In addition, our data showed that a lower ROS level was detected in stationary phase *S. aureus* than in exponential phase *S. aureus* (Appendix A). This is similar to a previous finding [43] and may indicate that the stationary phase cells are in a more metabolically inert state with less ROS being generated, and thus the corresponding capacity of the stationary phase cells to handle a sudden burst of ROS may be suboptimal compared to the actively growing exponential phase cells. Studies showed that the defense mechanism of bacteria is activated during exposure to exogenous H_2_O_2_ [44]. In our data, it is shown that 100 mM of H_2_O_2_ caused a larger increase in ROS detection in stationary phase bacteria than in exponential phase bacteria (Appendix A) and reduced bacterial survival in stationary phase *S. aureus* but not in exponential phase *S. aureus* (Appendix A). This indicated that stationary phase *S. aureus* has lower capacity in scavenging ROS, resulting in increased killing. The lower antioxidant capacity in stationary phase bacteria may account for the differential effect observed for SA-558 alone in stationary phase and exponential phase *S. aureus*. Taken together, SA-558 increased the ATP level and ROS production in stationary phase *S. aureus* and overwhelms the bacterial antioxidant capacity, which is lower compared to exponential phase *S. aureus*.

Antibiotic tolerance is implicated in chronic and difficult-to-treat infections in a clinical setting [45]. In this study, we established an in vivo model of *S. aureus* persistent infections with the ST239 IIIAH strain. A stationary phase culture treated by ciprofloxacin was characterized as the representative form of persistent *S. aureus* culture (Figure 5) as it fulfills the definition of ‘persisters’, which are a subset of the population that survive exposure to bactericidal antibiotic concentrations [8]. In Figure 5c, it is shown that ciprofloxacin, at 125 μg/mL, selects a proportion of the whole population (stationary phase culture). The ST239 IIIAH strain persists in mice even in the presence of vancomycin treatment and causes lethal infection at a lower inoculum than Mu3 (Figure 5d). Similar observations of SA-558 activity were attained in the ST239 IIIAH strain as those in the Mu3 strain (Appendix A). Vancomycin is the recommended treatment of MRSA infections [23], but it fails in 30–50% patients resulting in complicated bacteremia [24]. Studies [46,47] have shown that, similar to other conventional antibiotics, a metabolic shift in *S. aureus*, for example, downregulation of the TCA cycle, led to reduced susceptibility to being killed by vancomycin, contributing to tolerance. Targeting metabolic pathways determining vancomycin susceptibility facilitates bacterial killing [46,47]. We, thereby, reasoned that SA-558 might be active against antibiotic-tolerant bacteria by stimulating bacterial metabolism. As shown in Appendix A, SA-558 reduced abundance of stationary phase bacteria, which are tolerant to killing of antibiotics, including vancomycin in vitro. In the *S. aureus* persistent infection model, we showed that SA-558 alone is effective against the persistent form of *S. aureus*, of which vancomycin treatment fails to cause a significant reduction in bacterial abundance (Figure 1d and Figure 5d). Furthermore, our experimental data indicated that the combination of vancomycin and SA-558 caused significant reduction in bacterial abundance as compared to the vehicle control (*p* = 0.0013) and 4.5 mg/kg vancomycin treatment group (*p* = 0.02). In this study, we have shown that SA-558 is an effective therapeutic agent against antibiotic-tolerant *S. aureus*, of which even vancomycin fails to treat.

The vancomycin–SA-558 combination may be an effective strategy to be developed because bacteria are often present as a mixture of antibiotic-tolerant bacteria and susceptible bacteria; in the context of infection of the host, SA-558 and vancomycin can complement each other against bacteria at the infection sites. SA-558 targets antibiotic-tolerant bacteria while vancomycin, as the first-line antibiotic, is effective against susceptible bacteria. In addition, it is noteworthy that SA-558 is able to render a gentamicin-resistant strain (with MIC = 250–500 μg/mL) susceptible to gentamicin killing (Figure 1b,d, Appendix A). Furthermore, as SA-558 displays no toxicity in vitro and in vivo, it stands with high potential to be developed as an anti-persister therapeutic and will become a valuable part of our antibacterial armamentarium.

## 4. Materials and Methods

### 4.1. Bacterial Strains, Growth, and Plasmids

The bacterial strains used in this study were listed in Appendix A. All bacterial strains are maintained on Brain-Heart Infusion (BHI) agar (OXOID), incubated at 37 °C overnight and their antibiotic susceptibility was determined according to the guidelines suggested by the Clinical and Laboratory Standard Institute (CLSI) [48]. To vary the media pH, hydrochloric acid and sodium hydroxide were added for titration.

### 4.2. Antibiotic-Tolerant MRSA Killing Assay

*S. aureus* antibiotic-tolerant cells were isolated as previously described [6]. Briefly, *S. aureus* was cultured in LB to OD595 = 0.3 and diluted 1000-fold in LB, growing at 37 °C, with shaking overnight. The overnight culture was then washed with an equal volume of PBS. The pellet was resuspended in M9 minimal medium and adjusted to 10^7^ cfu/mL. The cultures were then exposed to treatment(s) of different antibiotics and SA-558 at 37 °C. Gentamicin (1 mg/mL), vancomycin (2 μg/mL), ciprofloxacin (125 μg/mL), oxacillin (4 mg/mL), and SA-558 (26.25 µg/mL) were used in the assay. At a specified timepoint, cultures in 200 μL were removed, serially diluted in PBS, and spot-plated onto BHI agar for CFU enumeration. Experiments were conducted in triplicate, and three repeats were performed for consistency.

### 4.3. Killing Kinetics Assay

An overnight culture of *S. aureus* Mu3 was diluted 1:100 into a fresh, cation-adjusted Mueller Hinton (CaMH) medium and cultured for 2 h to the exponential phase. The cultures were then treated with combination of SA-558 and gentamicin and incubated at 37 °C, with shaking at 250 rpm. At a specified timepoint, cultures, in 200 µL, were removed, serially diluted in Phosphate Buffered Saline (PBS), and spot-plated onto BHI agar for Colony Forming Unit (CFU) enumeration. Experiments were conducted in triplicate, and three repeats were performed for consistency.

To test the effect of CCCP eon SA-558 activity, the culture was pre-treated with 20 µM CCCP for 5 min at room temperature before being expose to a combinatorial treatment of SA-558 and gentamicin.

### 4.4. Membrane Potential Measurement (MPM)

The membrane potential of bacteria was monitored by anionic DiBAC4(3) dye, which enters depolarized cells and results in increased fluorescence. Hyperpolarization conversely prevents the entry of the dye and results in reduction in fluorescence. The MPM was performed as previously described [49]. *S. aureus* was harvested by centrifugation at 2400× *g* for 10 min, followed by washing twice and finally being resuspended in an equal volume of PBS. DiBAC4(3) (Thermofisher, Waltham, MA, USA), at a final concentration of 250 nM, was added to the suspension. The bacterial suspension, with DiBAC4(3) dye, was loaded into a 96-well black-opaque plate (SPL) in 200 µL per well, incubated at 37 °C with shaking in a SpectraMax Paradigm Multi-Mode Microplate Reader (Molecular Devices) for 30 min to allow equilibration, while monitoring DiBAC4(3) fluorescence (485 nm excitation, 535 nm emission). After equilibration, compounds or vehicle control was delivered in a volume of 1 µL to the 96-well plate, and DiBAC4(3) fluorescence monitoring was continued for another 30 min. The positive controls, which are 0.1% Triton X 100 and CCCP, were included in each assay.

### 4.5. Intracellular Gentamicin Concentration Quantification

Gentamicin and SA-558 concentrations in bacterial cells were assayed by LC-MS/MS. The sample preparation protocol was derived from Ebara et al.’s work [50]. Briefly, the pre-culture of *S. aureus* Mu3 was sub-cultured into CaMH medium at a ratio of 1:100 and shaken at 37 °C, 250 rpm for 2 h. The culture was then divided into 22 × 50 mL Falcon tubes, each with 20 mL of bacterial culture. Among the 22 tubes, 9 were treated with 15 µg/mL gentamicin, and another 9 were treated with the combination of 15 µg/mL gentamicin and 26.25 µg/mL SA-558. The remaining 4 tubes were left untreated for standard curve construction.

At 0.5, 1, and 2 h after treatment, bacterial cells were harvested by centrifugation at 2200× *g* for 10 min at 4 °C and washed with an equal volume of TES buffer [50 mM Tris-base, 5 mM EDTA, 50 mM NaCl, pH 7.5]. At this point, the bacterial suspension, in 1 mL, was removed for sample normalization. The cell pellet was subsequently resuspended in 1 mL TES buffer, and lysostaphin, 60 µg, was added, incubated at 37 °C for 30 min. The bacterial cells were further lysed by serial pulse sonication (30 s on and 30 s off), at 40% amplitude for 15 min. After that, the supernatant was collected after centrifugation 12,000× *g* for 10 min twice. For gentamicin samples, cell lysate, in 110 µL, was added to serial dilutions of gentamicin, 20 µL internal standard solutions (Kanamycin 100 µg/mL), and 50 µL 30% trichloroacetic acid (TCA) [50,51]. For SA-558 samples, 2 µL of cell lysate were added to serial dilutions of SA-558 work solutions and 20 µL internal standard solutions (M21, 3.85 µg/mL). The mixtures were centrifuged at 14,000 rpm for 10 min, and the supernatants were transferred to glass vials for MS analysis.

The gentamicin samples were analyzed using an Acquity UPLC system coupled to a Synapt G2-HDMS mass spectrometer system (Waters Corp., Milford, MA, USA). The chromatography was performed on a Waters ACQUITY UPLC HSS T3 column (100 × 2.1 mm; 1.8 µm). The mobile phase consisted of (A): 0.07% trifluoroacetic acid in water and (B) acetonitrile. The UPLC gradient program was applied as follows: 5% B (0 to 0.5 min), 5% B to 15% B (0.5 to 4.5 min), 15 to 98% B (4.5 to 5.0 min), 98% B (5.0 to 7.5 min), and 98 to 5% B (7.6 to 10 min). The LC-column temperature was maintained at 45 °C, and the injection volume was 5 µL. The sample manager was kept at 10 °C to avoid possible sample degradation. The mass spectral data were acquired in positive mode. The MS parameters such as capillary voltage, sampling cone voltage, and source offset were optimized and maintained at 2.5 kV, 60 V, and 60 V, respectively. The source and desolvation temperatures were maintained at 120 °C and 400 °C, respectively. Nitrogen gas was used for desolvation at a flow rate of 800 L/h.

The SA-558 samples were analyzed by the same instrument and analytical column as gentamicin. The mobile phase consisted of (A): 0.1% acetic acid in water and (B) acetonitrile. The UPLC gradient program was applied as follows: 40% B to 90%B (0 to 5 min), 90 to 100% B (5 to 5.1 min), 100% B (5.1 to 7.5 min), 100 to 40% B (7.5 to 7.6 min), and 40% B (7.6 to 10 min). The column and autosampler temperature were maintained at 45 °C and 10 °C, respectively [52]. The injection volume was 5 µL. The mass spectral data were acquired in negative mode, and other ionization source parameters were the same as gentamicin analysis.

### 4.6. Antibiotic Susceptibility Testing

The MICs of antibiotics were determined by the standard microdilution method recommended by the Clinical and Laboratory Standards Institute [48]. For SA-558, which dissolved in DMSO, serial dilutions were first prepared in a U-bottom polystyrene 96-well plate, and 1 μL of diluent was transferred to the assay plate containing the 50 μL medium. After that, 50 μL of bacterial suspension were added to the assay plate to achieve an inoculum of 5 × 10^5^ cfu/mL.

### 4.7. HPTS (8-Hydroxypyrene-1,3,6-trisulfonic acid) Assay

Base-pulse HPTS assays were conducted using liposomes loaded with pH-sensitive fluorescence dye HPTS (1 mM) to study the effect of compounds on pH dissipation [24]. To prepare HPTS-loaded liposome, asolectin from soybeans (Sigma-Aldrich, St. Louis, MO, USA) was dissolved in chloroform and evaporated in a round-bottom flask, and the formed lipid film was further dried under vacuum. The lipid film was resuspended in external salt solution (75 mM K_2_SO_4_, 10 mM HEPES, pH 6.8). The lipid suspension was then subjected to six freeze–thaw cycles and extruded 10 times through a 100 nm polycarbonate membrane. The unencapsulated HPTS was removed by size exclusion chromatography on a Sephadex G-25 column with HPTS-free external solution as the eluent.

Fluorescence measurement was performed with a F-2500 fluorescence spectrophotometer (Hitachi, Japan). For each measurement, the concentrated vesicle suspension was 10-fold diluted with external salt solution to obtain lipid suspension at 0.1 mM, in 2 mL. At 100 s, a base pulse (20 µL of 0.5 M KOH) was added to the vesicle suspension to generate a transmembrane pH gradient (pH_inside_ < pH_outside_). After that, the test compound was added, and the fluorescence ratio of HPTS (λex = 460 nm, λem = 510 nm divided by λex = 403 nm, λem = 510 nm) was recorded. At 500 s, 5% Triton X-100, in 40 µL, was added to destroy the pH gradient. The fractional fluorescence intensity (*I_f_*) was calculated using the following equation:
$$I_{f}=\frac{R_{t}-R_{0}}{R_{d}-R_{0}}$$
where *R_t_* is the fluorescence ratio at time t; *R*_0_ is the fluorescence ratio at time 0; and *R_d_* is the fluorescence ratio after detergent addition (end of experiment).

### 4.8. Intracellular pH Measurement

(i)Cloning of pGL-pHluorin plasmid

The pHluroin gene [53] was cloned under the control of SarA promoter [54] into the backbone of the *E. coli*/*S. aureus* shuttle vector derived from pGL [55]. The pHluorin gene sequence was codon-optimized for *S. aureus* based on Genbank sequence (AF058694.2) and synthesized by Genescript. The pHluroin gene was amplified by AccuPrime™ Taq DNA Polymerase, high fidelity (Thermofisher) using primer pairs pH-F/pH-R. The vector was prepared by inverse Polymerase Chain Reaction (PCR) using primer pairs RKP 2122/2123 with plasmid pGL-sarA as the DNA template. PCR products of the gene and vector were PCR purified by TaKaRa MiniBEST DNA Fragment Purification Kit and fused by ClonExpress II One Step Cloning Kit (Catalog: C112, Vazyme, Nanjing, China). The resulting plasmid pGL-pHluroin (Appendix A) was transformed into *E. coli* Top 10 using standard protocols. Plasmids of positive clones were confirmed by colony PCR and sequencing using RKP 2124/482. The plasmid of positive clones was isolated and transformed into *S. aureus* RN 4220 and subsequently the *S. aureus* strains of interest as previously described protocols [56]. The plasmid and primer sequences were included in Appendix A.

(ii)Fluorescence measurement

The *S. aureus* strains harboring pGL-pHluorin was pre-cultured overnight in Tryptic Soy Broth (TSB, Becton Dickinson, Sparks, MD, USA) with 10 μg/mL chloramphenicol, shaking at 37 °C, 250 rpm. The pre-culture was diluted at a ratio of 1:100 in 50 mL Falcon and grown until the exponential phase. The overnight culture was used as stationary phase bacteria. The bacterial culture was harvested by centrifugation at 12,000 rpm for 5 min at room temperature, washed with equal volume of PBS, and adjusted to OD 595 of 1.0 for fluorescence measurement. To measure ratiometric fluorescence of pHluorin, an excitation scan at 300–490 nm (emission at 510 nm) was performed using Hitachi F-4500 Fluorescence Spectrophotometer with a scan speed of 20 nm/s.

To measure the compound effect on intracellular pH, the compound was added into the bacterial suspension and immediately transferred to a quartz cuvette for fluorescence measurement. The excitation ratios (410/470 nm) were determined and interpolated for intracellular pH from the standard curve. CCCP, at 50 μM, was used as a positive control while untreated samples serve as the negative control. 

The standard curve of pHluorin was constructed by methods as described previously [57,58]. For each *S. aureus* strain tested, a standard curve was made for intracellular pH calibration.

### 4.9. ATP Measurement

Cellular ATP level was measured by using Cell Titer Glo reagent (Promega, Madison, WI, USA) as described previously [59] with some modifications. Three milliliter cultures grown to different growth phases were used for ATP measurement. To determine intracellular ATP concentration, triplicates of 10-µL supernatant samples were mixed with 10 µL Cell Titer Glo Reagent (reconstituted according to the manufacturer’s instructions) in wells of a 96-well white opaque flat-bottom plate, incubated at room temperature for 10 min. Luminescence was measured using a DTX 880 Multimode Detector (Beckman Coulter, Brea, CA, USA).

### 4.10. ROS Measurement

The ROS measurement protocol was adapted from Hoeksema, M. et al.’s study [60]. HPF (ThermoFisher Scientific, Waltham, MA, USA) and H2DCFDA (ThermoFisher Scientific), of which the fluorescence signal was proportional to the ROS, were used. The bacterial culture was diluted 1: 500 in LB medium and grown until OD_595_ of 0.6. Stationary phase bacteria were prepared by harvesting the overnight culture and washing and diluting it to 1:10 in M9 minimal medium. The fluorescent dye, at a final concentration of 5 µM, was added to the bacterial suspension and added to a 96-well black opaque plate (SPL) in a volume of 200 µL per well. Bacterial cells in different wells of the 96-well plates were treated with different antibiotic(s) and/or compounds, and the untreated control was included for comparison. The antibiotic treatments were performed in triplicate. HPF or H2DCFDA fluorescence (485 nm excitation, 535 nm emission) was measured in a DTX 880 Multimode Detector (Beckman Coulter, Brea, CA, USA) or SpectraMax Paradigm Multi-Mode Microplate Reader (Molecular Devices, San Jose, CA, USA) in a kinetic program consisting of 16 cycles of measurement at 30 min intervals at 37 °C, with shaking. OD_595_ of the bacterial culture was also recorded for normalization. Experiments related to H_2_O_2_ were included in Appendix A. 

### 4.11. In Vivo Studies

Animal experiments were performed according to protocols approved by the Laboratory Animal Unit, University of Hong Kong (CULATR no. 5097-19). Six- to eight-week-old female BALB/c mice were used in the in vivo studies.

(i)Determination of inoculum for murine persistent bacteremia infection model

To determine the culture which represents persisters, we prepared different forms of *S. aureus* ST239 IIIAH for infection, including a stationary phase culture, a ciprofloxacin-treated stationary phase culture, and an exponential phase culture. For the stationary phase culture, the strain was shaken at 37 °C, 250 rpm in BHI overnight. For the ciprofloxacin-treated stationary phase culture, the strain was grown in BHI for 4 h, and 4-fold MIC of ciprofloxacin was added to the cultures, which were then continued to be shaken at 37 °C, 250 rpm overnight. For the exponential phase culture, the overnight culture was sub-cultured in a BHI medium at a ratio of 1:100 and shaken at 37 °C, 250 rpm for 2 h.

Prior to the infection, the culture was harvested at 10,000× *g* for 10 min at 4 °C, washed twice with equal volume of PBS, and adjusted to 1 × 10^8^ cfu/mL by using PBS. CFU was enumerated by plating dilutions of bacterial suspension onto BHI agar plates. Bacterial suspension (1 × 10^7^ cfu) was delivered to mice by i.v. injection. Mice survival and the body weight change in mice were closely monitored and recorded. On 7 days post-infection, the mice were euthanized, and the kidneys were harvested for CFU enumeration.

(ii)SA-558 in vivo efficacy

An *S. aureus* murine persistent bacteremia model was used to evaluate SA-558 in vivo efficacy against persisters. The ciprofloxacin-treated stationary phase culture was prepared as described above and used for mice infection by i.v. The mice were monitored for 7 days to allow the development of persistent infections. On day 7 post-infection, the mice were randomized into different groups (*n* = 5–6 per group), and each group of mice was exposed to a different antibiotic/compound regimen, including (i) 4.5 mg/kg vancomycin alone, (ii) 15 mg/kg vancomycin alone, (iii) the combination of 4.5 mg/kg vancomycin and 2.63 mg/kg SA-558, (iv) 2.63 mg/kg SA-558 alone, and (v) the vehicle control, with 5% Tween 80 in PBS, supplemented with the same amount of DMSO. The drug treatments were administered twice a day by i.p. injection for 6 days. The mice were closely monitored until euthanasia. On day 13 post-infection, the mice were sacrificed, and the kidneys were aseptically harvested and homogenized for CFU enumeration. The results were repeated at least twice for consistency and pooled for statistical analysis. *p* values were determined by one-way ANOVA (* *p* < 0.05, ** *p* < 0.01 and *** *p* < 0.001).

## 5. Conclusions

The present study endeavored to investigate the ability of a small molecule compound, SA-558, in combating antibiotic tolerance in *S. aureus*. We found that SA-558 is effective against antibiotic-tolerant *S. aureus* in both in vitro and in vivo models. Mechanistic studies indicated that SA-558 targets the bacterial membrane, causing bacterial metabolic perturbation, for instance the proton motive force, resulting in lethality in the antibiotic-tolerant population. In addition to proton motive force perturbation, disruption of cellular homeostasis, induction of stress response, and ROS generation may contribute to SA-558 antimicrobial activities. This study highlights the importance of the bacterial membrane as a potential target for counteracting antibiotic tolerance. Our data indicated that SA-558 displayed no toxicity in vitro and in vivo. We envision that SA-558 stands with high potential as a therapeutic combating antibiotic tolerance.

## Figures and Tables

**Figure 1 ijms-24-06242-f001:**
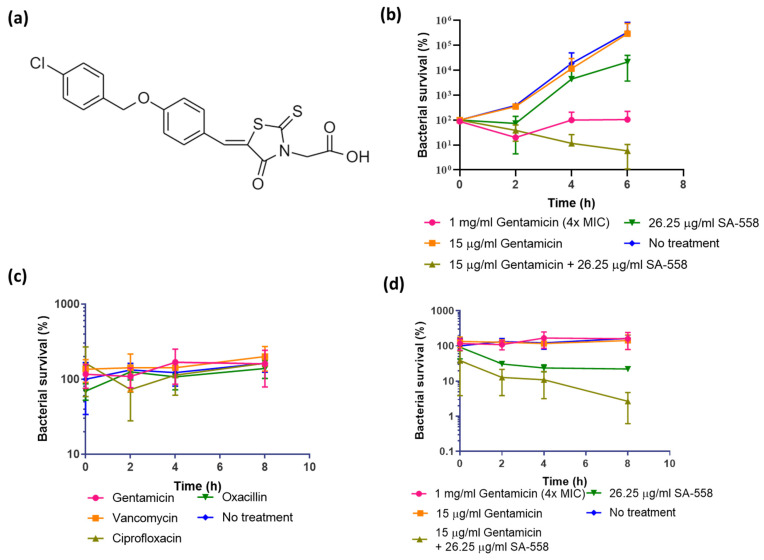
Antibacterial activity of SA-558 against *S. aureus* Mu3 strain in different growth phases. (**a**) Structure of SA-558; (**b**) Antimicrobial activity of SA-558 against exponential phase *S. aureus*; (**c**) Stationary phase *S. aureus* resisted killing by various antibiotics. Gentamicin (1 mg/mL), vancomycin (2 μg/mL), ciprofloxacin (125 μg/mL), oxacillin (4 mg/mL). Four-fold MIC of antibiotics were used in the assay; (**d**) Antimicrobial activity of SA-558 against stationary phase *S. aureus*. Means  ±  SD from three independent experiments are presented.

**Figure 2 ijms-24-06242-f002:**
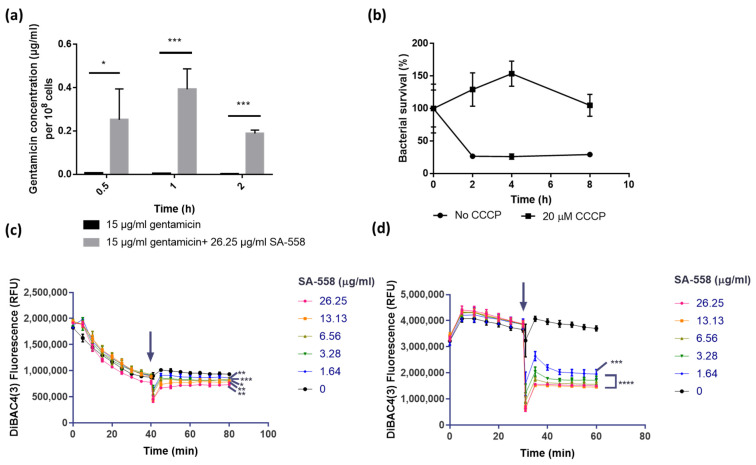
Effect of SA-558 on membrane potential and gentamicin uptake. (**a**) Total concentration of gentamicin inside *S. aureus* Mu3 after exposure to antibiotic/compound treatments. Statistical significance was determined by Student’s *t* test (* *p* < 0.05 and *** *p* < 0.001); (**b**) Time kill kinetics of combination of SA-558 and gentamicin on *S. aureus* Mu3 pre-treated with CCCP; (**c**,**d**) Effect of SA-558 at concentrations ranging from 26.25 to 1.64 μg/mL on membrane potential of the (**c**) exponential phase and (**d**) stationary phase *S. aureus* Mu3 strain. Change in DiBAC4(3) fluorescence over time was presented. The arrow indicated the time of compound addition. Statistical significance in comparison to untreated *S. aureus* was determined by one-way ANOVA (* *p* < 0.05, ** *p* < 0.01, *** *p* < 0.001 and **** *p* ≤ 0.0001).

**Figure 3 ijms-24-06242-f003:**
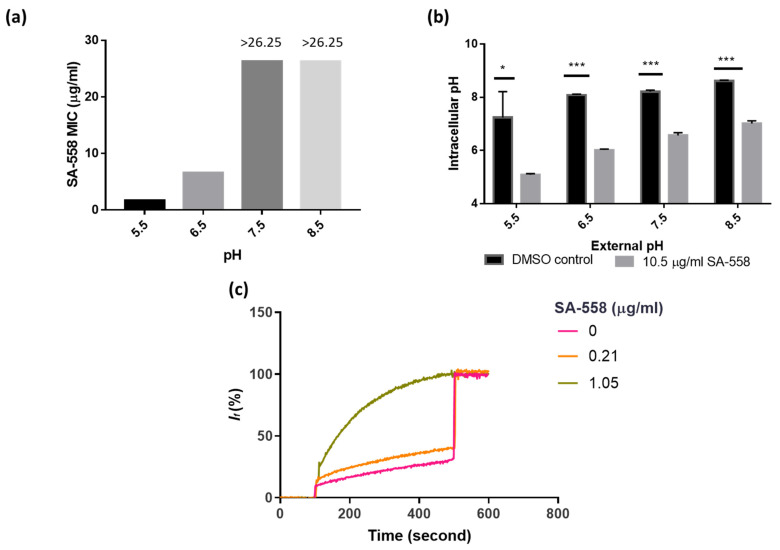
Effect of SA-558 on intracellular pH. (**a**) SA-558 MIC against *S. aureus* Mu3 in a function of media pH, from 5.5 to 8.5. (**b**) Effect of SA-558 on bacterial intracellular pH, at external pH ranging from 5.5 to 8.5. (**c**) SA-558 effect on HPTS-loaded liposome, at different concentrations. Statistical significance was determined by Student’s *t* test (* *p* < 0.05 and *** *p* < 0.001) Experiments were performed in three biological replicates. Bars represent mean ± SD (**b**).

**Figure 4 ijms-24-06242-f004:**
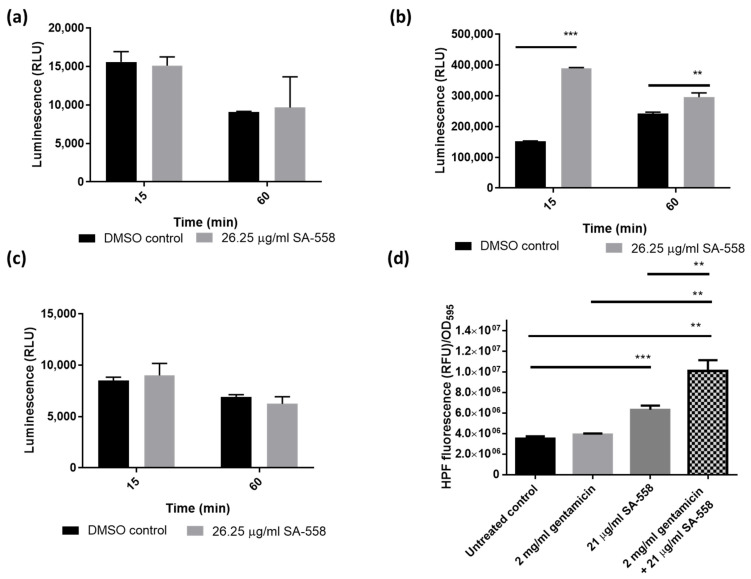
Effect of SA-558 on ATP and ROS level of bacterial cells. (**a**) Effect of SA-558 on bacterial cellular ATP level in early exponential phase; (**b**) Effect of SA-558 on bacterial cellular ATP level in early stationary phase; (**c**) Effect of SA-558 on cellular ATP level in Vero cells. Statistical significance was determined by Student’s *t* test (** *p* < 0.01 and *** *p* < 0.001); (**d**) Effect of SA-558 and/or gentamicin on ROS level of *S. aureus* Mu3, monitored by HPF normalized with OD_595_. (**e**) Effect of SA-558 and/or thiourea on ROS level of *S. aureus* Mu3, monitored by H2DCFDA normalized with OD_595_. (**f**) Effect of SA-558 and/or thiourea on bacterial survival of stationary phase *S. aureus* Mu3. Statistical significance was determined by one-way ANOVA (* *p* < 0.05, ** *p* < 0.01, *** *p* < 0.001 and **** *p* ≤ 0.0001). (**a**–**e**) The data are representative of three independent experiments. (**f**) Pooled means of three independent repeats were presented. Bars represent the mean ± SD.

**Figure 5 ijms-24-06242-f005:**
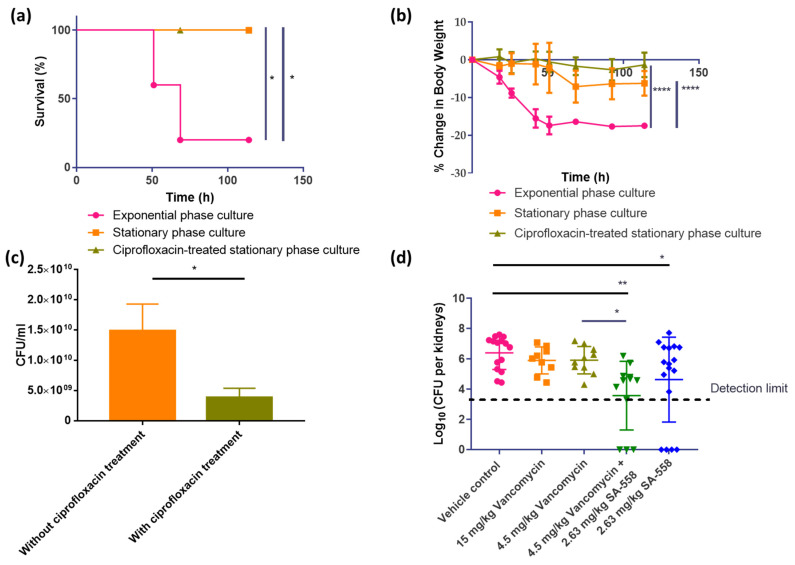
SA-558 in vivo efficacy in murine persistent bacteremia model. (**a**) Survival rates of mice. Mice (*n* = 5 per group) infected with different cultures of multidrug resistant *S. aureus* ST 239 IIIAH (1.0 × 10^7^ cfu) by i.v.. Statistical significance was determined by Log-rank (Mantel–Cox) test (* *p* < 0.05); (**b**) Body weight change in mice (*n* = 5 per group) infected with different cultures of *S. aureus* ST 239 IIIAH (1.0 × 10^7^ cfu) by i.v. Means ± SD were presented. Statistical significance was determined by ANOVA (mixed model) (**** *p* ≤ 0.0001); (**c**) Comparison of bacterial abundance in overnight culture of *S. aureus* ST239 IIIAH strain treated with or without ciprofloxacin. Means ± SD were presented, and statistical significance was determined by Student’s *t* test (* *p* < 0.05). (**d**) Comparison of bacterial load in kidneys of mice (*n* = 5–6 per group) receiving different treatments. The results were repeated at least twice for consistency and pooled for statistical analysis. Statistical significance was determined by one-way ANOVA. (* *p* < 0.05 and ** *p* < 0.01).

## Data Availability

Not applicable.

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
