# Peer review of "Identification of a Small Molecule Compound Active against Antibiotic-Tolerant Staphylococcus aureus by Boosting ATP Synthesis"

_ijms, 2023, doi:10.3390/ijms24076242_

Round 1

Reviewer 1 Report (Previous Reviewer 2)

I apologize for taking so much time to review the modified version of the manuscript. The authors provided answers to my questions and modified the text accordingly. In my opinion, the corrections made the text much clearer and easier to follow.

Few very minor comments and suggestions are left:

Lines 44-45: Consider changing the verbs “are” and “is” with “being”: It is a consequence of the biosynthetic processes that traditional antibiotics target for killing being significantly reduced [11] or antibiotic uptake being arrested as a result of low-energy state [12].

Line 61: I do not understand the using of “For which”. Shouldn’t it be “since” or “because”?

Line 130: I think the word “induction” is unnecessary, simply “proton motive force is required” looks fine. Consider removing it.

Line 163: “where ΔpH dominated” should probably be expanded like “where ΔpH component of the PMF dominated”

Line 202: preposition “with” is probably missing: interfere with ROS accumulation?

Line 206: maybe “explain the gentamicin tolerance” is better

Line 326: decreasess -> decreases (remove the double s)

Line 346: consider “ΔpH of proton…” -> “ΔpH component of proton…”

Past and present tenses are both used throughout the text. I think the usage of past tense is more appropriate when the results of the experiments are described. For example:

Line 94: caused

Line 103: were further tested

Line 167-170: maintained, decreased, varied, was more profound

Line 193: was not observed

Line 211: was suppressed by thiourea

Line 212: was reduced

Line 330: explored

Line 363: was reduced

I also wish to apologize if my comments might have appeared as overly critical and nitpicking, or were not always fomulated in an appropriately polite manner. 

Author Response

Response to reviewer

Response: Thank you for your comments and suggestions. We have revised the manuscript accordingly.

I apologize for taking so much time to review the modified version of the manuscript. The authors provided answers to my questions and modified the text accordingly. In my opinion, the corrections made the text much clearer and easier to follow.

 Few very minor comments and suggestions are left:

 Lines 44-45: Consider changing the verbs “are” and “is” with “being”: It is a consequence of the biosynthetic processes that traditional antibiotics target for killing being significantly reduced [11] or antibiotic uptake being arrested as a result of low-energy state [12].

Line 61: I do not understand the using of “For which”. Shouldn’t it be “since” or “because”?

Line 130: I think the word “induction” is unnecessary, simply “proton motive force is required” looks fine. Consider removing it.

Line 163: “where ΔpH dominated” should probably be expanded like “where ΔpH component of the PMF dominated”

Line 202: preposition “with” is probably missing: interfere with ROS accumulation?

Line 206: maybe “explain the gentamicin tolerance” is better

Line 326: decreasess -> decreases (remove the double s)

Line 346: consider “ΔpH of proton…” -> “ΔpH component of proton…”

 Past and present tenses are both used throughout the text. I think the usage of past tense is more appropriate when the results of the experiments are described. For example:

Line 94: caused

Line 103: were further tested

Line 167-170: maintained, decreased, varied, was more profound

Line 193: was not observed

Line 211: was suppressed by thiourea

Line 212: was reduced

Line 330: explored

Line 363: was reduced

 I also wish to apologize if my comments might have appeared as overly critical and nitpicking, or were not always fomulated in an appropriately polite manner. 

Reviewer 2 Report (New Reviewer)

The work is interesting and well planned with unique datasets, scientific techniques and graphical representations. The manuscript is also well written except a few minor corrections. The research finds a remedial compound against antibiotic tolerate bacteria Staphylococcus aureus.
1. I could not find the complete scientific name of the bacteria in the entire manuscript. Please write in full at its first mention in the text and also in the title.
2. Line 17: The sentence seems contradictory. Please rephrase it.
3. Line 397: Concentration of the antibiotics added?
4. Lines 449- 454: These sentences come under plagiarism. Please do the necessary. Also, check section 4.7; second para.
5. For Antibiotic(s) Susceptibility Testing was there any control kept with DMSO? Dimethyl sulfoxide has been observed to have bactericidal affect on Staphylococcus aureus.
6. The in vivo studies in methods need more clarity. The combination of the experimental groups and the methodology seems unstructured.

Author Response

Response to Reviewer

The work is interesting and well planned with unique datasets, scientific techniques and graphical representations. The manuscript is also well written except a few minor corrections. The research finds a remedial compound against antibiotic tolerate bacteria Staphylococcus aureus.
1. I could not find the complete scientific name of the bacteria in the entire manuscript. Please write in full at its first mention in the text and also in the title.

Response: Thank you for your comment. The full name of ‘Staphylococcus aureus’ has been added in the title (Line 2-3) and text (Line 21).
2. Line 17: The sentence seems contradictory. Please rephrase it.

Response: Thank you for your comment. The sentence has been revised (Line 17).
3. Line 397: Concentration of the antibiotics added?

Response: Concentrations of antibiotics have been added (Line 407-408).
4. Lines 449- 454: These sentences come under plagiarism. Please do the necessary. Also, check section 4.7; second para.

Response: The relevant text has been paraphrased (Line 469-475) and (Line 502-508).

5. For Antibiotic(s) Susceptibility Testing was there any control kept with DMSO? Dimethyl sulfoxide has been observed to have bactericidal affect on Staphylococcus aureus.

Response: In each assay, DMSO control (1% DMSO) was included. We observed no effect on bacterial growth with 1% DMSO tested.
6. The in vivo studies in methods need more clarity. The combination of the experimental groups and the methodology seems unstructured.

Response: Thank you for your comment. The text has been modified.

This manuscript is a resubmission of an earlier submission. The following is a list of the peer review reports and author responses from that submission.

Round 1

Reviewer 1 Report

GENERAL COMMENTS

This is a potentially important paper, but it is not yet ready for publication. Many of the problems would be corrected by a better knowledge of persistence, tolerance, and antimicrobial-mediated lethality. As a starting point, I would suggest careful reading of all papers on these subjects by JJ Colins, Xilin Zhao, and Naomi Balaban. The generality of the developing concepts can be found in Shee 2022 AAC, which deals with M. tuberculosis. There are several reviews that may be helpful: Balaban 2019 (Nat Rev Microbiol), Amit Singh 2022 (Frontiers), and Drlica 2020 or 2021 (Expert Reviews in Antimicrobial …).

Experimental problems are listed below. I emphasize that it is important to clarify the role of SA-558 in drug accumulation and killing (see comment line 211). This may be difficult with amino glycosides; perhaps some experiments with a potent fluoroquinolone could be used to establish principles that could then be applied to amino glycosides.

The English usage is generally good, although the proper use of articles (a, an, the) and verb-subject agreement could be improved. The reference list requires much more attention to detail.

SPECIFIC COMMENTS

Line 32   it is important to establish that you examining persisters. You need to show that they are a small proportion of the population. Otherwise, you may be measuring tolerance, which involve the entire population, which is not persistence.

Line 39 bad better as poor

Lines 43-50 the term efficacy is too vague, because persistence, as a form of tolerance, affects killing, not inhibition of growth (as per definitions of Tomasz 1986 and Balaban 2019). Many readers confuse tolerance with a change in MIC which would be a part of the term efficacy. Thus, use of efficacy may undermine author credibility. Line 69 has the same problem with the word potentiation: too vague, suggesting lack of author knowledge of the literature.

Line 75 Figure 1bc. These panels show killing of cells in stationary phase, but the methods describe killing only for exponentially growing cells.  Two things are important when considering stationary-phase cells. First, they are not suitable models for persistence by directly measuring antimicrobial-mediated killing, because antimicrobial lethality generally requires metabolism and thus nutrients. The inability to kill stationary-phase cells can be due to failure to achieve the initial drug-mediated lesion (usually measured as lack of growth inhibition which cannot be measured with non-growing cells) and/or it can be due to a form of tolerance. Failure to experimentally separate these two phenomena makes interpretation of stationary-phase experiments ambiguous. Second, death can occur during and after antimicrobial treatment (during growth of colonies on agar after drug removal). The latter reveals ROS effects. Failure to accumulate ROS would generate tolerance. The problem with aminoglycosides is thar they cannot be easily washed out of cells. Thus, it is likely that the death on plates is due to SA-558 increasing the effective concentration of gentamicin. Increasing concentration affects the primary lesion formation and subsequent death. If you do not separate the two, which is exceptionally difficult with aminoglycosides, interpretation about persisters and tolerance being affected by SA-558 is ambiguous.  These considerations do not mean you are wrong about SA-558, but they undermine author credibility and therefore the strength of your conclusioins.

Line 77-80. You claim that SA-558 alone at 26 ug/ml does not reduce survival of exponentially growing cells. But you should also examing higher concentrations of SA-558 because exponentially growing cells have a higher membrane potential than stationary-phase cells and thus require more SA-558 to test for a perturbation.

Line 80 selectively … you claim activity against isolated persisters, but I do not see the lack of activity against non-persisters, as required by the term selectivity.  Point out where these data are. 

Line 102 activity … this term includes inhibition of growth. You need to show that no MIC change occurred to make your interpretation unambiguous. This is the same problem as lines  43-50 but more serious because you need the MIC data. Note that changes in uptake will change MIC, further confusing your interpretation.Line 140… where is the evidence for selectivity?

Line 136 instantaneously … where are the data? 15 min is hardly instantaneous. Overinterpretation undermines author credibility.

Figure 4bd. Since SA-558 increases membrane potential of stationary but not exponential cells, it is possible that fluorescent probes for ATP and ROS assays accumulate to higher levels in stationary cells in the presence of SA-558, thus showing higher fluorescence signals that are not necessarily due to increased ROS and ATP. It seems counterintuitive that ATP levels would be rapidly increased at stationary phase due to absence of nutrients and active metabolism. Thus, additional controls are required.

Fig 5a I do not see the triangles in the panel

Line 211 If persistence is a subcategory of tolerance, as per Balaban 2019, then antibiotic accumulation, which affects MIC, cannot be claimed as responsible for persistence. The issue is that killing follows a drug-target interaction (MIC) as ROS accumulation and death occur. Tolerance and persistence are defined as failure to kill, not as change in drug uptake. Changes in uptake make conclusions ambiguous (but not necessarily wrong). If you are correct that accumulation is affected by your compound, you should show this by MIC changes. Then you need to rephrase the paper to establish that you are really looking at the killing process – or that you do not know the contribution to killing per se vs accumulation of agent. These effects are more clearly seen with fluoroquinolones where blocking growth (MIC) is clearly distinct from killing.

Reference list is not carefully formulated. You should not list authors by only one initial without a name.

Reviewer 2 Report

The manuscript entitled “Identification of a Small-molecule Compound Active against S. aureus persisters by Boosting ATP synthesis” describes the identification of a small molecule compound SA-558 and characterization of its activity towards S. aureus. SA-558 appears to be able to target persister cells and synergize with traditional antibiotics. The manuscript is well written and easy to read, the experiments seem to have been conducted thoroughly with appropriate controls. However, there are several issues that seem confusing and that, in my point of view, require clarification.

1. My main concern is the explanation that authors provide for the observed SA-558 activity. As I understood, the authors suggest that SA-558 dissipates (decreases) the PMF of bacterial cells (especially at low pH when the PMF is mostly due to the excess protons outside of the cell), which results in the increase of ATP synthesis, ROS generation, and gentamicin uptake. However, PMF is required for ATP synthesis (except in the case of glycolysis), and it is not clear how PMF dissipation could result in the increase of ATP synthesis. Furthermore, high PMF could be expected to be associated with actively growing and respiring cells, not persisters. Persisters possess low metabolic activity and thus do not require as high PMF as active cells. If SA-558 decreased the PMF, it would be expected to target exponential cells rather than persisters. Section 2.1 demonstrates that SA-558 in combination with gentamicin kills both stationary and exponential cells with approximately the same efficiency, rather than specifically targets either cell type. Moreover, CCCP dissipates the PMF, and the activity of SA-558 was abolished by CCCP. This means SA-558 requires the PMF for its action, rather than disrupts it. If SA-558 dissipated the PMF, CCCP would have the same effect as SA-558. Instead, CCCP abolishes the activity of SA-558. The authors provide some explanation in lines 250-260. I must admit, however, I could not follow the logic in the explanation, and found it confusing.

2. Another important point is the pH dependence of SA-558 action. The authors show that SA-558 is more active at lower pH and connect it with the different nature of the PMF at different pH. A much more straightforward and plausible explanation is simply the difference in SA-558 charge at different pH. As evident from Figure 1, SA-558 has a carboxylic group. It can be expected that at lower pH this group will be protonated and SA-558 will be neutral, while at higher pH the carboxylic group will lose the proton and SA-558 will bear a net negative charge of -1. Uncharged SA-558 at low pH would more efficiently penetrate bacterial membranes (charged molecules do not penetrate the membranes) and act upon its target within bacterial cell (whichever this target might be, possibly the membrane itself). This action would lead to cell damage, which in turn could cause some other effects like the inability to maintain intracellular pH and increased ROS generation. In fact, increased ROS generation is observed with most antibiotics.

3. The strains chosen for this study appear to be resistant to gentamicin (MICs provided in supplementary are 250-500 ug/mL). It is not clear why a gentamicin-resistant strain was chosen to study the potentiation of gentamicin towards persister cells. If the strain is intrinsically resistant to gentamicin, how would one discriminate between persisters and normal cells? As can be seen in Figure 1, 1 mg/mL gentamicin (a huge concentration!) is equally ineffective against both stationary and exponential cells, and equally potentiated by SA-558. Moreover, when gentamicin is used in combination with SA-558, its concentration is much much lower (15 ug/mL). Thus, it appears that SA-558 can actually make a gentamicin-resistant strain susceptible. If this is true, it is an even more interesting result than potentiation of gentamicin activity towards persisters.

4. Gentamicin is used for in vitro experiments, and one experiment shows its increased uptake in the presence of SA-558. Increased uptake of aminoglycosides in general is often described as a mechanism of potentiating their action towards persister cells. However, vancomycin is used in vivo. The mechanisms of action of these two antibiotics are very different (vancomycin is not an aminoglycoside but a glycopeptide). Importantly, vancomycin acts from the outside (it blocks the synthesis of peptidoglylcan), and thus does not require uptake to kill bacteria. The choice of antibiotics should be clarified.

Minor remarks:

The phrase “electroneutral transfer of charge” is confusing. I have never seen such a term and could not find it in Google. If the authors have introduced this term in the present manuscript, it should be explained (what exactly does it mean?).

Line 25, S. aureus is not italicized

Line 49, “at” should probably be removed

Line 54, “of which” – it looks like something is missing here

Line 74, “were found” should probably be replaced with something like “are known”. “Were found” can be understood as if it was found in the present study.

Line 77, “when combined” should probably be replaced with “when SA-558 was combined”

Line 88, ”at” should probably be removed

Line 114, “in a function” -> “as a function”

Line 159, “culture by ciprofloxacin treatment” should probably be replaced with “culture treated by ciprofloxacin”

Line 170, I would replace “being” with “as”

Line 198, “at” should probably be removed

Line 277, “were” -> “are”

Line 307, in “washing for twice”, “for” should probably be removed

Line 337-338, “exposed to centrifugation” should probably be replaced with “centrifuged”

Line 472, “supplemented same” - probably “with” is missing (“supplemented with the same”)

To sum up, I like the experimental level at which this work has been performed and I do want to see it published. However, the explanation of the observed effects provided by the authors does not seem compelling. I strongly suggest putting less emphasis on the PMF disruption by SA-558 in the Results and Discussion sections and considering alternative theories for SA-558 mechanism of action.

Round 2

Reviewer 1 Report

GENERAL COMMENTS

This is a potentially important paper, but it is not yet ready for publication. Many of the problems would be corrected by a better knowledge of persistence, tolerance, and antimicrobial-mediated lethality. As a starting point, I would suggest careful reading of all papers on these subjects by JJ Colins, Xilin Zhao, and Naomi Balaban. The generality of the developing concepts can be found in Shee 2022 AAC, which deals with M. tuberculosis. There are several reviews that may be helpful: Balaban 2019 (Nat Rev Microbiol), Amit Singh 2022 (Frontiers), and Drlica 2020 or 2021 (Expert Reviews in Antimicrobial …).

Experimental problems are listed below. I emphasize that it is important to clarify the role of SA-558 in drug accumulation and killing (see comment line 211). This may be difficult with amino glycosides; perhaps some experiments with a potent fluoroquinolone could be used to establish principles that could then be applied to amino glycosides.

The English usage is generally good, although the proper use of articles (a, an, the) and verb-subject agreement could be improved. The reference list requires much more attention to detail.

SPECIFIC COMMENTS

Line 32   it is important to establish that you examining persisters. You need to show that they are a small proportion of the population. Otherwise, you may be measuring tolerance, which involve the entire population, which is not persistence.

Line 39 bad better as poor

Lines 43-50 the term efficacy is too vague, because persistence, as a form of tolerance, affects killing, not inhibition of growth (as per definitions of Tomasz 1986 and Balaban 2019). Many readers confuse tolerance with a change in MIC which would be a part of the term efficacy. Thus, use of efficacy may undermine author credibility. Line 69 has the same problem with the word potentiation: too vague, suggesting lack of author knowledge of the literature.

Line 75 Figure 1bc. These panels show killing of cells in stationary phase, but the methods describe killing only for exponentially growing cells.  Two things are important when considering stationary-phase cells. First, they are not suitable models for persistence by directly measuring antimicrobial-mediated killing, because antimicrobial lethality generally requires metabolism and thus nutrients. The inability to kill stationary-phase cells can be due to failure to achieve the initial drug-mediated lesion (usually measured as lack of growth inhibition which cannot be measured with non-growing cells) and/or it can be due to a form of tolerance. Failure to experimentally separate these two phenomena makes interpretation of stationary-phase experiments ambiguous. Second, death can occur during and after antimicrobial treatment (during growth of colonies on agar after drug removal). The latter reveals ROS effects. Failure to accumulate ROS would generate tolerance. The problem with aminoglycosides is thar they cannot be easily washed out of cells. Thus, it is likely that the death on plates is due to SA-558 increasing the effective concentration of gentamicin. Increasing concentration affects the primary lesion formation and subsequent death. If you do not separate the two, which is exceptionally difficult with aminoglycosides, interpretation about persisters and tolerance being affected by SA-558 is ambiguous.  These considerations do not mean you are wrong about SA-558, but they undermine author credibility and therefore the strength of your conclusioins.

Line 77-80. You claim that SA-558 alone at 26 ug/ml does not reduce survival of exponentially growing cells. But you should also examing higher concentrations of SA-558 because exponentially growing cells have a higher membrane potential than stationary-phase cells and thus require more SA-558 to test for a perturbation.

Line 80 selectively … you claim activity against isolated persisters, but I do not see the lack of activity against non-persisters, as required by the term selectivity.  Point out where these data are. 

Line 102 activity … this term includes inhibition of growth. You need to show that no MIC change occurred to make your interpretation unambiguous. This is the same problem as lines  43-50 but more serious because you need the MIC data. Note that changes in uptake will change MIC, further confusing your interpretation.Line 140… where is the evidence for selectivity?

Line 136 instantaneously … where are the data? 15 min is hardly instantaneous. Overinterpretation undermines author credibility.

Figure 4bd. Since SA-558 increases membrane potential of stationary but not exponential cells, it is possible that fluorescent probes for ATP and ROS assays accumulate to higher levels in stationary cells in the presence of SA-558, thus showing higher fluorescence signals that are not necessarily due to increased ROS and ATP. It seems counterintuitive that ATP levels would be rapidly increased at stationary phase due to absence of nutrients and active metabolism. Thus, additional controls are required.

Fig 5a I do not see the triangles in the panel

Line 211 If persistence is a subcategory of tolerance, as per Balaban 2019, then antibiotic accumulation, which affects MIC, cannot be claimed as responsible for persistence. The issue is that killing follows a drug-target interaction (MIC) as ROS accumulation and death occur. Tolerance and persistence are defined as failure to kill, not as change in drug uptake. Changes in uptake make conclusions ambiguous (but not necessarily wrong). If you are correct that accumulation is affected by your compound, you should show this by MIC changes. Then you need to rephrase the paper to establish that you are really looking at the killing process – or that you do not know the contribution to killing per se vs accumulation of agent. These effects are more clearly seen with fluoroquinolones where blocking growth (MIC) is clearly distinct from killing.

Reference list is not carefully formulated. You should not list authors by only one initial without a name.

Reviewer 2 Report

I apologize for the delayed submission of my review report.

The authors provided answers to my questions and modified the manuscript accordingly. However, upon careful reading of the revised version, I realized the problem was not so much the explanations the authors provided for the observed effects, but the places in the manuscript where these explanations first appear, and the general style of the manuscript. Here are some examples:

Line 87-90: the authors say they identified the compounds that act on metabolism. However, in this section (2.1) they only identify a compound that potentiate gentamicin against a gentamicin-resistant strain. It might or might not act on the metabolism – this is not known from the experiments in this section, it only becomes clear later in the text. The authors present this conclusion here, but the reader does not yet know where it comes from, and this creates confusion. The reader thinks: why do they give this conclusion, it is not supported by the data!

Section 2.2 is named “SA-558 targets at proton motif force”. However, in this section, there are no experiments that demonstrate it! The experiments in this section only show that SA-558 increases the membrane polarization, and that this effect is important for the tobramycin-potentiating activity (because CCCP abolishes it). However, the section does not show what the target of SA-558 is. Only later in the text the authors show that SA-558 targets the membrane and equilibrates the pH (in the liposomes experiment). A better title for this section would thus be “SA-558 effect is mediated by the increase in membrane polarization” or something like that.

Section 2.3. Again, the section name is not exactly what you show in the section. The section is titled “SA-558 dissipated proton gradient component of proton motive force”. However, the experiments in this section show that the activity of SA-558 is pH dependent and that it can equilibrate the pH on the two sides of the membrane (liposomal or bacterial). Of course, if one combines this information with the rest of the experiments, one might assume that such conclusion (about the dissipation of the PMF) is possible. But it is not directly demonstrated in this section. Such conclusions should be part of the Discussion.

Lines 167-168: “This ascertains that SA-558 targets at ΔpH component of proton motive force” – the experiments described in this section certainly do not ascertain it. They do ascertain that SA-558 activity is pH-dependent and that SA-558 decreases the intracellular pH. Nothing more. This observation suggests that such equilibration of intracellular pH with the external pH might be responsible for the observed hyperpolarization of the membrane that compensates for the dissipation of the ΔpH component of the PMF (as the authors suppose). But it definitely does not ascertain it. In any case, such suggestion should either be moved to the end of the section (after you’ve shown that SA-558 equilibrates the pH inside the liposomes) or moved to the Discussion.

Lines 176-179: the same problem again – the authors give a conclusion when it is not warranted. It is completely unclear at this point why the metals are mentioned. There have been no experiments in the manuscript that attempted at measuring the metal ions. There is no discussion explaining the metal ions. The authors just present them with no clear reason at all.

Line 361: again, the metals are not explained. What makes you think SA-558 mediates metal transport? It is not explained at all, whether it is true or not.

This problem – that the authors present the conclusions before the data that actually support them – deteriorates the logical coherence of the text and leads to a lot of confusion. 

Several other issues:

Line 116: wi?

Figure 2. The panel (d) description is still called (c) in the figure caption.

Lines 172-173: the pH in the bacteria have been moved to the previous paragraph. This phrase should probably be moved as well.

Lines 315-317: isn’t it vice versa, at low pH ΔpH predominates? It contradicts your further words (two lines later you say ΔpH predominates at low pH) and even your idea on the mechanism of SA-558, which is supposed to dissipate the ΔpH component at low pH. But if, as you say here, ΔpH collapses at low pH, how would SA-558 dissipate it at low pH?

Line 326: as far as I know, CCCP dissipates all PMF, not just Δψ – it just moves protons across the membrane.

Finally, a couple of remarks regarding the answer to the first question of my previous review:

As I understand, the dissipation of the ΔpH component of the proton motive force basically means that the cell can no longer maintain the intracellular pH, meaning it equalizes with the external pH, meaning the concentrations of protons on both sides of the membrane become the same. If so, then what SA-558 does is the disruption of the pH homeostasis (as authors themselves suggest in the answer to the last comment). Then these two effects (“dissipation of the ΔpH component of the PMF” and “disruption of the pH homeostasis”) are the same effect called two different names. The latter name (“disruption of the pH homeostasis”) is, at least to my taste, more straightforward and easier to grasp. I would use it (or a similar phrase) to refer to this effect.

Considering the third part of the answer, I do not understand how it proves your point. If SA-558 increases the ATP production and converts the persisters to the normal growing state, then the preferential action towards stationary cells should only be seen for the combination of SA-558 and gentamicin (because gentamicin would then be able to kill these no-longer-persisters). And SA-558 alone should not demonstrate any difference in the effect towards exponential and stationary cells. But we see the opposite, the differential effect is observed for SA-558 alone, but not for the SA-558/gentamicin combination.